# Automatic beam optimization method for scanning electron microscopy based on electron beam Kernel estimation
Yunje Cho , Junghee Cho, Jonghyeok Park, Jeonghyun Wang, Seunggyo Jeong , Jubok Lee, Yun Hwang, Jiwoong Kim, Jeongwoo Yu, Heesu Chung, Hyenok Park, Subong Shon, Taeyong Jo, Myungjun Lee & Kwangrak Kim ✉

Scanning Electron Microscopy (SEM) leverages electron wavelengths for nanoscale imaging, necessitating precise parameter adjustments like focus, stigmator, and aperture alignment. However, traditional methods depend on skilled personnel and are time-consuming. Existing auto-focus and auto-stigmation techniques face challenges due to interdependent nature of these parameters and sample diversity. We propose a beam kernel estimation method to independently optimize SEM parameters, regardless of sample variations. Our approach untangles parameter influences, enabling concurrent optimization of focus, stigmator x, y, and aperture-align x, y. It achieves robust performance, with average errors of 1.00 μm for focus, 0.30% for stigmators, and 0.79% for aperture alignment, surpassing sharpness-based approach with its average errors of 6.42 μm for focus and 2.32% for stigmators and lacking in aperture-align capabilities. Our approach addresses SEM parameter interplay via blind deconvolution, facilitating rapid and automated optimization, thereby enhancing precision, efficiency, and applicability across scientific and industrial domains.

The Scanning Electron Microscope (SEM) is a pivotal tool in the analysis of material microstructures, utilizing the short wavelengths of electrons to observe objects at the nanoscale. This capability to employ electrons with wavelengths of less than a nanometer (nm) allows SEM to transcend the limits of conventional optical microscopy, enabling high-resolution observations of nanoscale structures. Consequently, SEM has become an indispensable instrument in a diverse array of research fields, including biomedical science, material science, and nanoscience.

In the semiconductor industry, SEM has found extensive application in metrology and inspection during the device manufacturing process[1]. It is routinely employed for critical tasks such as critical dimension (CD) measurement and overlay assessment, which evaluate the precision of the patterning process following lithography or etching procedures. Additionally, as device-impacting particles and pattern sizes shrink to the tens of nanometers scale, SEM's role in defect and particle detection has grown[2], largely because ensuring the reliability of defect detection with optical inspection tools has become increasingly challenging. In addition to its traditional applications, SEM has recently been harnessed for more advanced techniques, including three-dimensional measurements using high-acceleration beams and the analysis of electrical properties through voltage contrast[3-6]. However, compared to traditional optical microscopes, SEM presents distinct operational challenges. Most optical equipment is designed to focus on robust and reliable optical systems. During operation, controlling the focus according to the measurement location is typically sufficient. In contrast, SEM requires a more comprehensive consideration of various factors and parameters to shape and control the electron beam, achieving high-resolution images. This complexity is compounded by the variable nature of the electron source and electromagnetic lenses, or the observation conditions like beam current and acceleration voltage. As a result, SEM equipment, particularly in research facilities, often requires skilled personnel for its operation. Even in industrial settings, ensuring the stability and reliability of SEM is a major challenge.

Key operational requirements for SEM in semiconductor manufacturing include accuracy, reliability, throughput, and automation[7-9] capability. The accuracy and reliability of the SEM equipment, crucial for its resolution, depend on the precise control of focus and aberration. This precision enables the consistent acquisition of sharp and correct images. Figure 1 exemplifies the effects of beam parameter variations on image quality. It displays SEM images under different conditions: **a** shows a sharp image without any blurriness, **b** an image with out-of-focus but without astigmatism, **c** an image on-focus but with astigmatism, and **d** an image out-of-focus and with astigmatism. These examples highlight how deviations

Metrology and Inspection Equipment R&D Team, Mechatronics Research, Samsung Electronics Co., Ltd., 1-1 Samsungjeonja-ro, hwaseong-si, Gyeonggi-do 18848, Republic of Korea. ✉e-mail: kwangrak.kim@samsung.com

**Fig. 1 | Blurriness in scanning electron microscopy images due to beam parameter variations.** Image blurriness due to out of focus, astigmatism a ~ d are images under following conditions: a A sharp image without any blurriness. b An image of out-of-focus, but with astigmatism. c An image of on-focus but with astigmatism. d An image of out-of-focus but with astigmatism.

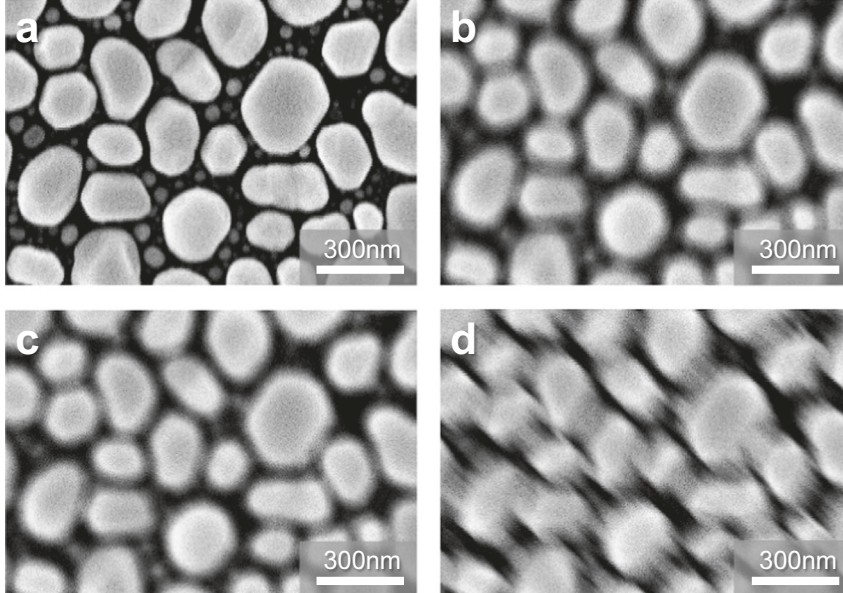

from optimal conditions, such as out-of-focus or stigmator aberration, lead to blurred or directionally blurred images. Such image quality issues can result in measurement inaccuracies during CD measurement and cause false positives or failures in defect inspection. Given the trend toward sub-nanometer scale patterning precision in advanced device processes, the need for technology that can precisely optimize the electron beam is paramount. Moreover, this optimization may need to be performed frequently due to changes in the beam over time, different beam conditions used, and the loading/unloading of wafers. Therefore, establishing a methodology for automatic and rapid beam optimization is essential.

Beam optimization in SEM is challenging for two primary reasons. First, the hardware configuration required to form and control the beam is complex, necessitating the simultaneous optimization of multiple parameters. Second, there is no direct method to verify the shape of the beam; its form must be inferred from the captured images. For these reasons, a profound understanding of the hardware configuration and the mechanisms for electron beam control, as well as the image generation process in SEM, is vital for effective beam optimization. While the specific configuration of SEM[4,5] apparatus may vary by manufacturer, a typical setup comprises an electron gun, an electron-optical column, and a main chamber. The column houses several modules crucial for focusing and controlling the electrons emitted from the gun. Important components such as the condenser lens, objective lens, stigmator, and aperture are integral to the concentration of the beam, correction of astigmatism, and positional alignment. Precise control of three main parameters—focus, stigmator, and aperture alignment—is vital to obtain sharp and accurate images. Misalignment of these parameters can lead to blurred images, thereby leading to inaccuracies in measurements.

Various automatic methodologies have been proposed and explored within the academic realm for calibrating the parameters of the electron beam, especially focus and stigmator[10,11]. Notable research has been conducted on the concept of autofocus, particularly with image sharpness-based approaches[12–24]. These studies incorporate the assumption that the optimal combination of parameters yields the sharpest image. Various methods exist for assessing sharpness, such as evaluating the gradient magnitude of images, analyzing the distribution of magnitudes in the Fourier domain, and detecting edges. Using a sharpness-based approach to sweep parameters, as done for focus adjustment, results in a one-dimensional optimization process. However, as the number of parameters requiring adjustment increases, the search space becomes multidimensional. This complexity makes it challenging to find the optimal point using a full search approach.

Particularly in SEM, the strong interdependencies among beam parameters often result in numerous false optimal points from a sharpness perspective, leading to frequent optimization failures. More advanced methods involve frequency domain analysis[25–30], where ellipse fitting in the frequency domain enables the prediction of blur size and direction based on the size and orientation of the ellipse. These methods enable the control of three parameters: focus, stigmator (x, y).

However, the frequency domain approach faces limitations in differentiating between mixed directionalities of sample patterns and blur. Consequently, this approach can exhibit substantial error when dealing with patterns exhibiting clear directionalities. Another research trend involves designing score functions to estimate defocus and astigmatism using correlation or contrast transfer functions[31–35]. While these methods have their merits, they are vulnerable to noise in an image and do not measure scores robustly for severely blurred images. Notably, recent studies have employed deep learning[36–44] to calculate sharpness by training relationships between images and scores assigned by humans. These papers adopt a convolutional neural network to process electron microscopy images. The inference time is near-instant and shows satisfying results. However, this approach controls only focus or three parameters (focus, stigmator x, and y) and, being data-dependent, requires a substantial amount of data and time for training a neural network. Additionally, it needs recalibration for new types of samples. Other methods have been developed using TEM/STEM to target specific aberrations of a particular equipment and resolve them by utilizing additional features such as tilt. However, this is not a universally applicable technique for equipment like SEM[45–51].

In this paper, we introduce a methodology that optimizes five essential SEM parameters—focus, stigmator (x, y), and aperture alignment (x, y)—regardless of the pattern's orientation. Differing from conventional methods that often encounter false optima due to the close interdependence of beam parameters, our approach uses beam kernel estimation to dismantle these interdependencies. We have formulated a blind deconvolution model, building upon previous studies[52,53]. This model simplifies the process of acquiring sharp, high-resolution SEM images, making it more accessible and efficient for both trained and non-trained personnel. We anticipate that our methodology will contribute Substantially not only to the automation of semiconductor analysis in high-volume manufacturing but also to enhancing the general usability of SEM in various scientific and industrial applications.

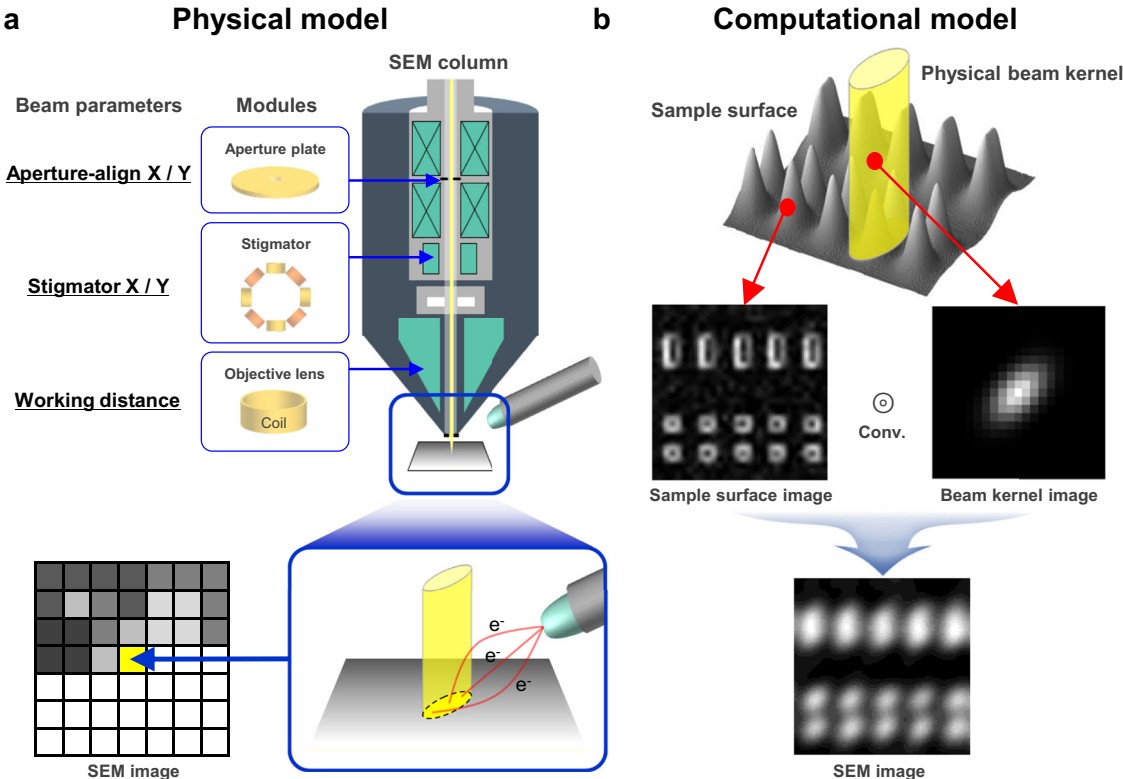

**Fig. 2 | Physical and computational model of scanning electron microscopy (SEM) image acquisition. a** The physical model of the SEM image acquisition process, showing how an image is acquired through surface scans influenced by beam parameters such as aperture-align x, y, stigmator x, y, and working distance.

**b** The computational model simulating the sampling of surface scans with a beam kernel through convolution[55] between a sample surface image and a beam kernel image.

## Results

In this paper, we introduce a methodology that leverages the estimation of beam kernels for the automatic optimization of SEM beam parameters, thereby eliminating the need for manual hardware adjustments. Our approach unfolds in a methodical manner across five key subsections: Firstly, the SEM image acquisition model is presented, establishing a foundational understanding of how SEM images are generated and influenced by various beam parameters. This model is essential for comprehending the subsequent steps in beam optimization. Then, we delve into the intricacies of the beam kernel, exploring its characteristics and how it correlates with the beam parameters. Understanding the interplay between the beam kernel and the parameters that dictate SEM imaging quality is pivotal in this context. Following this, the core of our methodology involves estimating the beam kernel by unraveling the inverse model of the image acquisition process. This step is critical for deducing the beam conditions from the observed images, enabling a more direct and effective optimization process. Subsequently, we explain how to extract beam parameter correction values from the estimated beam kernel. This process translates the theoretical model into practical adjustments that can be applied to the SEM, targeting the optimal settings for each parameter. Finally, we demonstrate how the proposed method concurrently optimizes five critical beam parameters—focus, stigmator x, y, and aperture-align x, y. The effectiveness of this comprehensive optimization is evaluated by analyzing error dispersion, which calculates error distance for each case and subsequently provides the mean and deviation across them. This assessment validates our methodology and highlights its potential to noticeably enhance the precision and efficiency of SEM imaging.

### Image acquisition in SEM: beam Kernel—object convolution

In this study, we interpret the physical model of the SEM imaging process by employing a computational model of image convolution for the numerical

analysis of electron beam control. Figure 2a illustrates the physical model of the image acquisition process in an SEM, demonstrating the influence of beam parameters on SEM image acquisition. This process can be divided into two phases: the formation of an electron beam within the column and the determination of each pixel's value for constructing a 2D image. The shape of the scanning electron beam is determined by various hardware devices installed within the column[54], such as the emitter, aperture, stigmator, and objective lens. The cross-sectional shape of the electron beam irradiated on the sample's surface (focused in a circular or elliptical shape) correlates with the volume of interaction with the sample material, and the intensity of signals, such as secondary electrons, determines each pixel's value in the image.

In this physical model, the optimization problem of the electron beam arises as the shape of the beam cannot be directly measured but must be inferred through acquired images. The resulting signal, as the electron beam irradiates the surface area, manifests as a weighted sum according to the beam kernel's shape, filling each image pixel. Consequently, the beam kernel operates analogously to an optical blur kernel or a point spread function. Discretizing an acquired SEM image conceptualizes it as the convolution[55] of a surface pattern image with the beam kernel image. Figure 2b illustrates the computational model interpreting the SEM acquisition process as a convolution between these two images: a sample surface image and a beam kernel image. Modeling this process as a convolution problem enables the numerical analysis and optimization of the physical elements of the beam kernel, as detailed in the following section.

### Interdependence between beam Kernel and beam control parameters

The pursuit of optimal quality in SEM imaging is intrinsically linked to achieving the smallest and least deformed beam kernel. A smaller beam kernel enhances resolution, while deformation in a beam kernel leads to

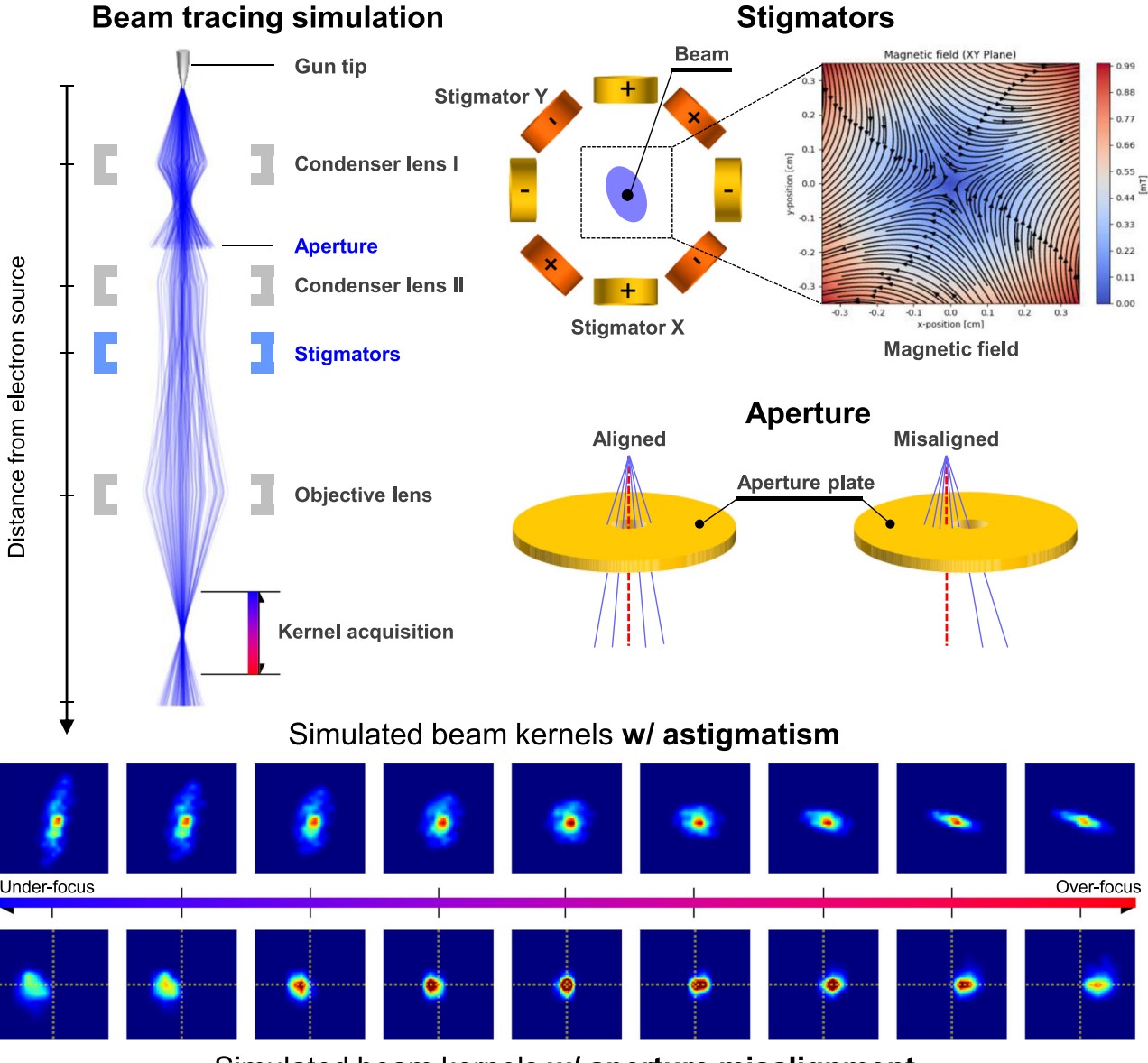

**Fig. 3 | Beam tracing simulation in Scanning Electron Microscopy (SEM): impact of stigmator and aperture on beam kernel.** This figure illustrates a Monte Carlo simulation of electron beam tracing in SEM, highlighting the effects of stigmator and aperture adjustments on the beam kernel. The simulation incorporates the influence of magnetic fields from the stigmator, condenser lens, and objective lens on the electron beam's path. The aperture module, represented as a fixed-size hole, allows for vertical and horizontal alignment adjustments. The bottom two rows display a series of beam kernel images, demonstrating variations of astigmatism, and aperture misalignment based on working distance.

blurring in specific image directions. The formation of the beam kernel is influenced by factors such as the magnetic fields created by the objective lens and stigmator, and the alignment of the aperture hole. Essentially, it's the beam parameters controlling these components that determine the beam kernel's characteristics.

To elucidate the relationship between the beam kernel and these parameters, we conducted beam tracing simulations to observe the kernel's shape under various parameter settings. The validity of these simulations was confirmed by comparing simulated kernels with images acquired from actual SEM equipment.

Figure 3 presents the setup of our beam tracing simulation. Electrons emitted from the gun pass through key components such as the condenser lenses, aperture, stigmator, and objective lens, each playing a crucial role in shaping the beam. The simulation includes only the critical components influencing the beam path. The images on the right side of Fig. 3 depict the

magnetic field XY cross-sections for the stigmators. The Lorentz force causes the electrons along the beam path to be deflected, with the cumulative path changes of individual electrons defining the beam kernel shape. Aperture alignment, which adjusts the holes' positions along the beam path, can cause a tilt in the beam axis when misaligned. This tilt results in variations in the beam kernel position depending on the working distance. The simulations slice the beam cross-section at regular intervals beneath the objective lens to verify the kernel's shape and position, as illustrated below Fig. 3. The images reveal that the direction and position of the beam kernel shift from under focus to over focus.

In SEM, core beam parameters like focus, stigmator, and aperture-align substantially influence the formation of the beam kernel. The objective lens modulates the magnetic field magnitude by adjusting the coil current, thereby regulating the beam's convergence distance. The stigmator, with its eight coils arranged in facing pairs, performs the roles of stigmator X + , X-,

## a Stigmator consistency between simulated beam kernels and SEM images

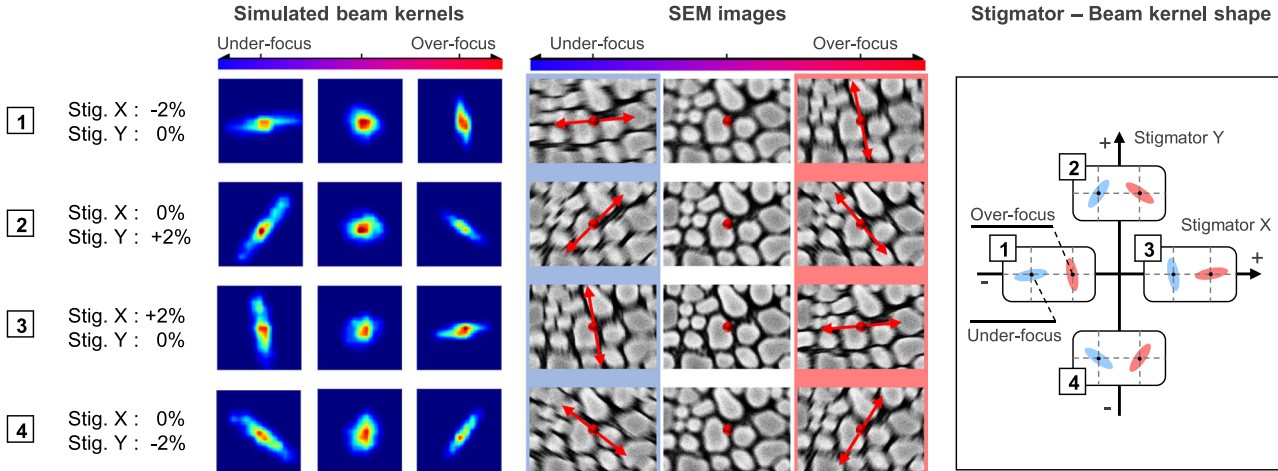

## b Aperture-align consistency between simulated beam kernels and real images

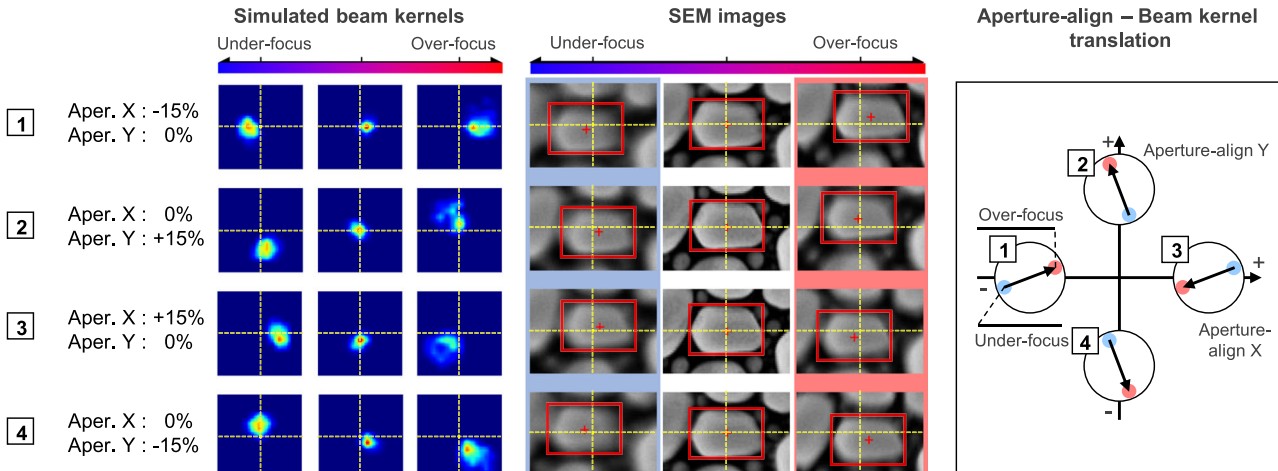

**Fig. 4 | Simulated and experimental validation of stigmator and aperture alignment in Scanning Electron Microscopy (SEM).** This figure showcases a comparison between beam kernel trends from simulations and actual SEM images under varying conditions. **a** displays both simulated beam kernels and corresponding SEM images under different stigmator conditions, with a schematic on the right illustrating the variations in beam kernel shape based on these conditions. The direction of beam kernel deformation varies with stigmator x and y adjustments, observable as perpendicular orientations between under-focus and over-focus conditions. **b** Presents the simulated beam kernels and experimental images for different aperture alignment scenarios. The movement of specific structures within the red box closely mirrors the simulated beam kernel translation. The right-side schematic depicts how the beam kernel's translation is influenced by the aperture alignment, indicating a shift in direction when transitioning from under-focus to over-focus.

Y +, and Y-. Adjustments to these parameters can transform the elliptical beam kernel into a circular shape. Aperture alignment, on the other hand, collimates and aligns the beam, with misalignment introducing a translation in the beam kernel, necessitating correction.

Figure 4a investigates the beam kernel shape changes according to stigmator conditions, compared against specific condition images obtained experimentally. This comparison reveals how deviations in stigmator X and Y affect the beam kernel size, inducing a blur in the image. Changes in the stigmator cause the beam kernel to deform into an elliptical shape in specific directions, with X and Y inducing vertical/horizontal and diagonal deformations, respectively. Additionally, for the same stigmator values, the major axis orientation of the elliptical beam kernel is perpendicular between under-focused and over-focused conditions. Figure 4b demonstrates how aperture alignment translation affects the beam kernel, with trends consistent in both experimental and simulation results. The direction of movement of specific structures within a red bounding box in the experiment matches the simulated beam kernel's movement direction. This movement, determined by aperture alignment conditions, indicates how aperture misalignment causes

the kernel to shift directionally, posing challenges in stigmator adjustment and potentially leading to sample positioning inaccuracies. It's noteworthy that X and Y in the aperture parameter do not represent the same directions in the imaging plane due to the Lorentz force-induced rotation.

In summary, incorrect states of the stigmator and aperture-align lead to beam kernel deformation, thereby blurring SEM images. Consequently, our method ascertains the beam kernel state, enabling us to reverse engineer these parameters to yield high-resolution images.

### Blind deconvolution approach for electron beam Kernel estimation

As established in the previous section, determining the beam kernel is vital for understanding beam parameters, especially given their dependency on the working distance. However, directly obtaining the beam kernel within an SEM is not feasible. To address this challenge, we have developed an indirect method to extract the beam kernel from acquired images. Figure 5 illustrates the optimization process of the beam kernel using a blind deconvolution formulation[42,43,56,57].

**Fig. 5 | Iterative reconstruction process of beam Kernels by solving inverse model.** The inverse model, reversing the computational process, aims to derive multiple beam kernels and a sample surface image from Scanning Electron Microscopy (SEM) images taken at varying working distances. The iterative process for blind deconvolution is depicted, showcasing how beam kernels and a sample surface image evolve with each iteration. Starting from symmetric Gaussian kernels and a random noise image, the beam kernels' directionality becomes more distinct, and patterns in the sample surface image become increasingly visible as the iterations progress.

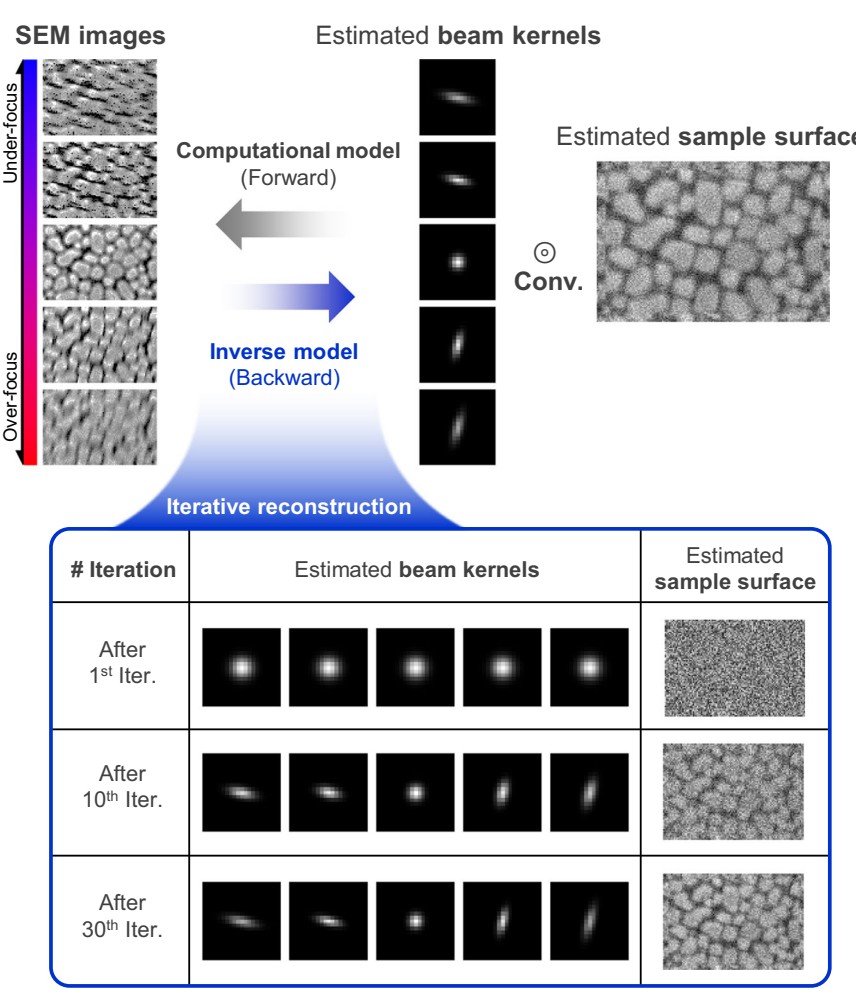

Building on the computational model discussed in the first section, where SEM images result from the convolution of a beam kernel image with a surface image, we address the inverse problem. Given that only SEM images are available, we formulated a mathematical model using blind deconvolution techniques to estimate both the surface image and the kernel when both are unknown. Blind deconvolution employs strong constraints, such as Total Variation (TV)[42,43], to facilitate this estimation process.

Considering the alterations in beam parameters from under-focus to over-focus, as shown in Fig. 4, we designed an objective function to leverage focus sweep images. This function aids in extracting beam kernels along with information about their blur direction and translation. In the blind deconvolution problem, imposing intense constraints is essential due to the limited data available. Our constraints include the assumption that focus sweep images share the same surface image but with different kernels, the presumption that the correct surface image is sharp (TV regularization), and the beam kernel has Gaussian shape. The imposition of these constraints markedly reduces the number of parameters requiring estimation, thereby enhancing convergence stability. With our method, only a limited set of parameters are estimated, specifically three values for constructing the symmetric covariance matrix of the Gaussian kernel for each working distance, along with two additional positional values for their translations. The following objective function, shown in Eq. (1), is formulated to be minimized by our blind deconvolution algorithm:

$$F = \min_{u, \Sigma_1, \cdots, \Sigma_N, v_1, \cdots, v_N} \frac{1}{N} \sum_{i=1}^{N} ||k_{\Sigma_i} * u(x + v_i) - f_i||_2^2 + \lambda ||\nabla u||_1^1 \quad (1)$$

In the equation, $N$ is the number of sweep images, $f_i$ represents the $i$th image from the focus-sweep image set obtained through experiment, $u$ is the

estimated surface image, $x$ is the positional vector, $v_i$ is the $i$th translation vector, $k_{\Sigma_i}$ is the $i$th estimated beam kernel with 2D Gaussian covariance matrix $\Sigma_i$, and $\lambda$ is balancing parameter of TV term.

The blind deconvolution process involves learning a series of beam kernels and a surface image, starting from an isotropic Gaussian kernel and a random noise image. The kernel and surface image are iteratively refined to reduce the error between the obtained SEM image and the convolution result. Figure 5 demonstrates this iterative process, where the estimated beam kernel gradually gains directionality, and the estimated surface image increasingly reflects the pattern of the sample.

## Feature-based beam parameter correction method

In this section, we discuss the update process for beam parameters using the estimated beam kernels obtained through the blind deconvolution technique. This technique facilitates the extraction of beam kernels for each working distance, along with a common surface image shared by sweep images. The characteristics of these kernels, such as size, shape, and displacement direction, are closely linked to the beam parameters. Drawing on these characteristics, we have developed a method to correct key parameters, namely the stigmator, aperture alignment, and focus, for acquiring a correct image. Figure 6 illustrates the process of extracting compensatory directions for these key parameters from the acquired beam kernels.

The values of stigmator X and Y are related to the beam kernel's shape. During the transition from under-focus to over-focus, if the stigmator remains at its optimal value, the beam kernel size diminishes until reaching a minimum at the exact focus point, then expands again in over-focus. Conversely, with non-optimal stigmator values, the kernel shape transitions from a skewed ellipse to a circle at focus and then to a vertically skewed ellipse, perpendicular to its shape in under-focus.

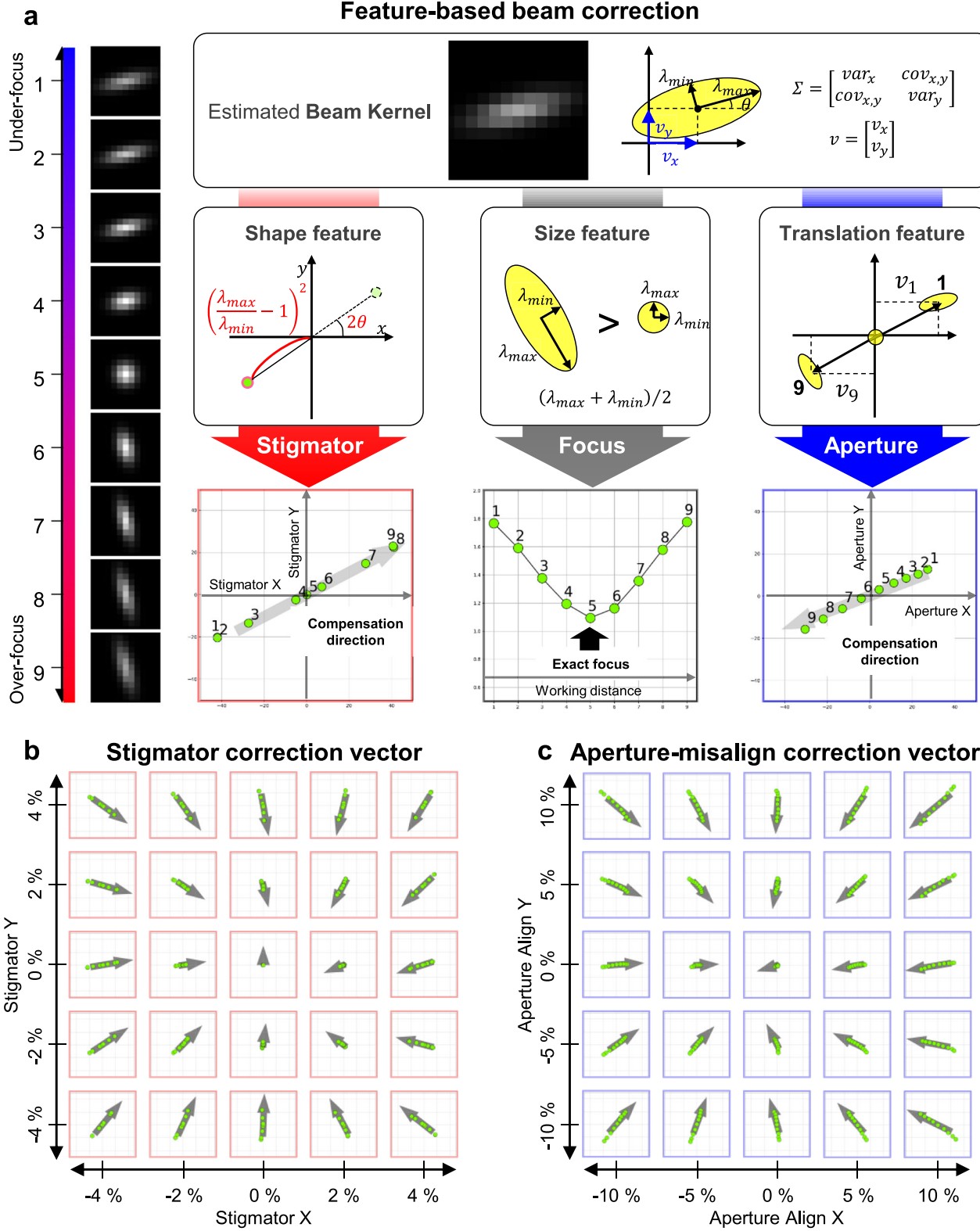

**Fig. 6 | Feature-based beam correction method. a** Extraction of shape, size, and translation features from Gaussian-shaped beam kernels, used for correcting stigmator, focus, and aperture alignment. A set of estimated beam kernel images from under-focus to over-focus is utilized for correction. **b** Stigmator compensation vectors calculated for various stigmator conditions. **c** Aperture-align compensation vectors determined for different aperture-align conditions. The working distance is swept within ±12 μm from the on-focus reference, using a Gold on Carbon sample.

We characterize these shape changes of the kernel from under-focus to over-focus. Initially, we extract the major and minor axes of the elliptical kernel. The length ratio of these axes and the angle between the major axis and the x-axis are quantified. At perfect focus, where the kernel is circular, we set the size feature to zero, employing the formula $\left(\lambda_{max}/\lambda_{min} - 1\right)^2$. The feature angle is set to twice the angle between the major axis and the x-axis, accounting for the ellipse orientation. This doubling aligns the major axis angles for under and over-focus linearly at 180 degrees. Feature points

for all transitioning kernels are then mapped onto Cartesian coordinates, illustrating their linear alignment. From these features, we determine the compensation direction, using the standard deviation of the feature points as the compensation magnitude. This method ensures that as the stigmator nears its optimum, the kernel shape converges towards a circle, centering near the origin and thus reducing the feature size.

For aperture misalignment, where the electron beam does not irradiate the sample vertically, we observe a specific shift in the beam kernel's position with focus change. This shift direction corresponds to the aperture's XY plane misalignment. We use image translation rather than kernel translation for computational efficiency. The translation vectors from under-focus to over-focus are depicted in Cartesian coordinates, with compensatory vectors calculated to capture translation vector trends. The direction of the first principal component vector calculated by Principal Component Analysis (PCA)[58] determines the compensatory vectors' direction, and their magnitudes are proportional to the standard deviation of the translation vectors' Euclidean distance distribution.

Focus optimization occurs at the point where the beam kernel is smallest. We compute the kernel size as the average of the ellipse's major and minor axis lengths for each focus-dependent kernel, selecting the smallest value as the optimal focus point.

To validate the correction vector direction, we obtained focus sweep images of Gold on Carbon (GoC) samples under various stigmator and aperture alignment conditions. Applying our method to these conditions, we extracted correction vectors, as shown in Fig. 6b, c. The direction of these vectors indicates the optimal direction, and their magnitude decreases as they approach the optimal value, for both stigmator and aperture alignment. The x and y coordinates represent relative values, with optimal values set at 0% for both axes.

## Performance evaluation

The proposed approach, integrated into an in-house software program for real-time control, continuously acquires images while iteratively updating five parameters to achieve final convergence. The specifics of this process and the operating program are further detailed in the methods section. To assess the performance of the proposed method, we measured the accuracy of the beam parameters at final convergence compared to ground truth (GT) values manually adjusted by trained personnel. The accuracy was determined as the average of the Euclidean distance errors in both the 1D case, such as the working distance, and the 2D case, including the stigmator and aperture alignment.

The experiments began with all beam parameters set markedly off from the GT, at levels where sample patterns were difficult to discern. Two initial parameter settings were tested: one with fixed aperture-align value and working distance but varied stigmator values, and another with fixed stigmator values and working distance but varied aperture alignment values. Figure 7a illustrates the process from initial status to optimized status, showing convergence of stigmator, aperture-align, and working distance to positions near the GT within 15 iterations. Figure 7b, c demonstrates the convergence of all parameters near the GT for various initial stigmator or aperture-align conditions. The average distances from the converged points to the GT are as follows: focus, 0.29 μm (**b**) and 0.28 μm (**c**); stigmator, 0.34% (**b**) and 0.08% (**c**); and aperture-align, 0.17% (**b**) and 0.19% (**c**).

The proposed method's performance was compared with conventional sharpness-based methods using auto-correlation[31,32] in Fig. 8. These methods were applied as auto-focus followed by auto-stigmation (AF → AS) and auto-stigmation followed by auto-focus (AS → AF). Experiments were conducted on Gold on Carbon (GoC) and two semiconductor logic device samples under four conditions: out-of-focus (Case I), astigmatism (Case II), out-of-focus with astigmatism (Cases III). When aperture-align x, y deviates from the optimum points, as well as the preceding three parameters (CASE IV), conventional correction methods may be ineffective. However, all five parameters can be accurately adjusted simultaneously using the proposed method. It shows notably lower error levels (focus: 1.00 μm, stigmator: 0.30%, aperture-align: 0.79%) for all GoC, Logic I, and II

samples. Figure 9 compares final result images calibrated with each method. Conventional methods show some pattern emergence in Cases I and II, but in Case IV, where all parameters are misaligned, the image patterns are entirely absent. However, the proposed algorithm maintains image quality similar to the GT for all prepared samples, even with misaligned focus, stigmator, and aperture.

The conventional method's limitations stem from the interdependence among beam parameters. In contrast, our method, by introducing and estimating beam kernels, dismantles these interdependencies. This approach has two advantages: it allows simultaneous optimization even with multiple misaligned parameters and operates reliably across various samples due to its pattern-independent beam kernel extraction.

## Discussion

This paper has introduced a method for generating and manipulating beam kernels to facilitate the acquisition of high-resolution SEM images through automated beam parameter adjustments. Initially, we explained the SEM image acquisition process as a convolution of surface patterns and beam kernels, incorporating this concept into a computational model. The relationship between the beam kernel and beam parameters was substantiated through a combination of simulations and experimental validations. We proposed an inverse model of this computational framework to extract the beam kernel, followed by a method to estimate beam parameters using the extracted kernel. Performance evaluations confirmed that our method consistently produces sharp, high-resolution images, even when all beam parameters are misaligned.

The proposed method offers three important advantages. Firstly, it elucidates the relationship between beam parameters and image quality by introducing the concept of a beam kernel. This understanding is pivotal for automated beam parameter correction. Secondly, a key strength of our method lies in its ability to perform pattern-independent operations, distinguishing it from Fourier domain techniques and deep learning approaches. While these conventional methods often face challenges with pattern differentiation or necessitate training data similar to the sample pattern, our method excels in its versatility and adaptability, making it highly suitable for a wide range of general-purpose applications. Finally, the simultaneous optimization of five parameters—stigmator X, Y, aperture-align X, Y, and working distance—is a notable achievement. This multi-parameter optimization accelerates image acquisition preparations, a crucial benefit when dealing with constantly changing samples.

The potential applications of this method extend across various domains utilizing SEM technology. Its significance is particularly notable in analytical laboratories with SEMs where sample types frequently change, necessitating constant adjustment of conditions. Similarly, in FIB-SEM[59] systems, where state variations require regular condition readjustments after each milling step, our method could be invaluable. While the focus has primarily been on the semiconductor field, the versatility of this technique suggests its utility in other areas, such as biology, where it could substantially reduce the time required for specimen preparation and observation.

## Methods
### Experimental setup

The experiments conducted utilized a range of samples including a reference Gold on Carbon (GoC) sample and two types of Logic device semiconductor samples. The GoC sample, featuring gold particles deposited onto a carbon substrate, exhibits diverse round-shaped, non-repetitive structures of various sizes. Logic I sample is characterized by repetitive square patterns along with larger structural elements, while Logic II includes various curved line patterns, such as horizontal and vertical lines.

A stand-alone Zeiss Gemini 460 SEM[60] equipped with the Gemini 2 Column and double condenser lenses was employed for these experiments. This SEM model is noted for maintaining low resolution degradation even under high beam current conditions. It houses critical components like the aperture, stigmator, and objective lens, which are controllable via joystick or API input. Although the SEM originally comes with Zeiss's SmartSEM[61]

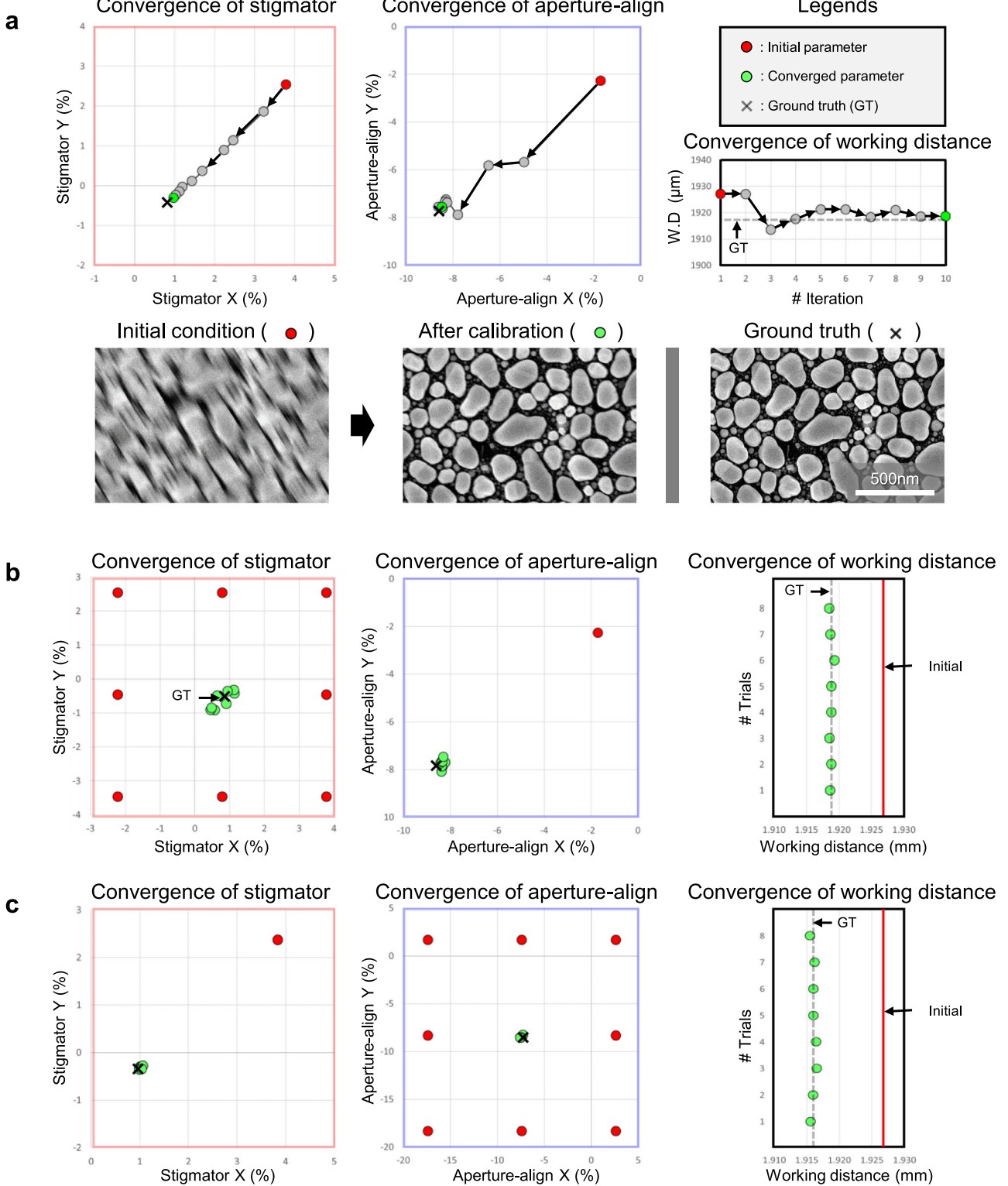

**Fig. 7 | Automatic beam optimization process and convergence of beam parameters. a** Graph showing the optimization trace from initial conditions (red circle) to final convergence (yellow-green circle), with the GT represented by the gray x-mark or dashed line. Images depict initial conditions, post-optimization, and at GT. **b** Convergence from varied initial stigmator positions with incorrect aperture-align and working distance. **c** Similar to (**b**), but starting with varied aperture-align and fixed stigmator and working distance.

software, a custom C#-based auto-recipe program was developed to interface with the API. This setup allows for sequence control and full automation of beam parameter optimization and image acquisition. For advanced computations, a processing PC connected to the SEM via HTTP communication is used. This PC is outfitted with a GeForce RTX 3090 graphics card, an Intel Xeon Gold 6234 CPU, and 256GB of RAM. It runs a Python-based Flask server that handles requests and images from the SEM PC, processes them, and returns the resulting correction values. The server

## a CASE I – Focus misaligned

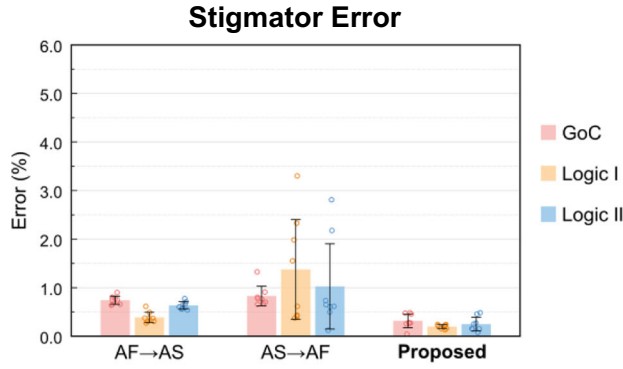

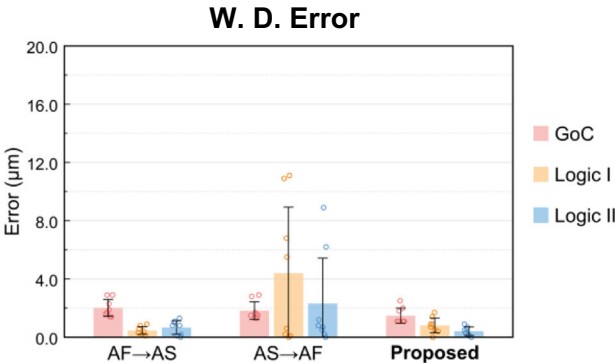

## b CASE II – Stigmator misaligned

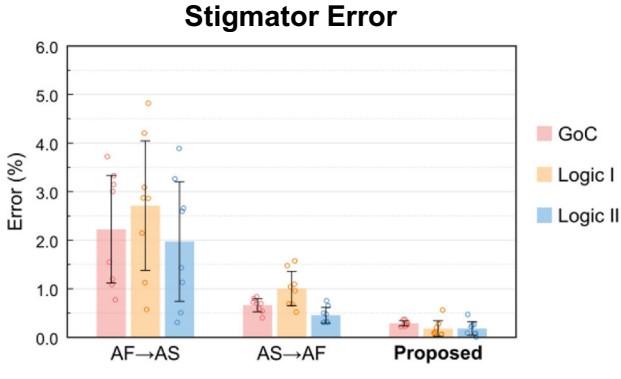

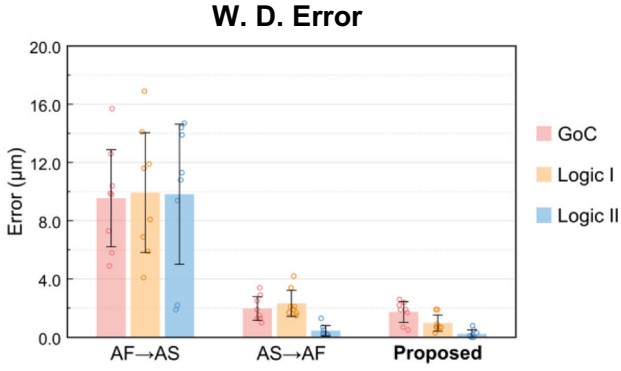

## c CASE III – Focus, Stigmator misaligned

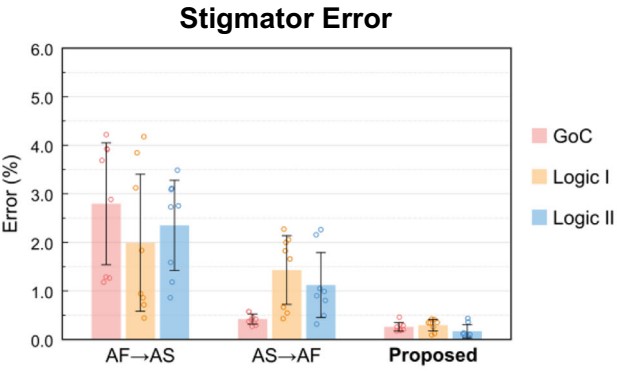

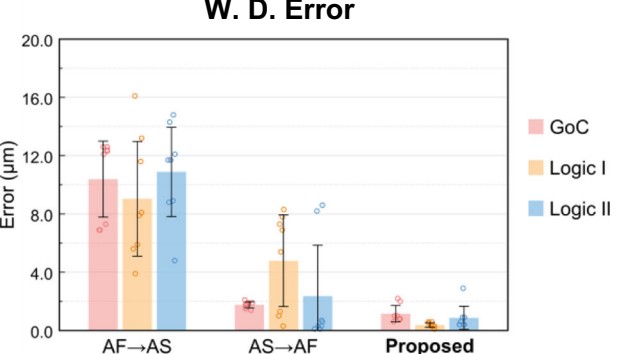

**Fig. 8 | Accuracy performance evaluation.** Performance evaluations for three cases (**a** CASE I: focus misaligned, **b** CASE II: stigmator misaligned, and **c** CASE III: both focus and stigmator misaligned) across three samples (GoC, Logic I, and II). Error dispersion for stigmator and working distance (W. D.) is shown, along with comparisons made between sharpness based methods (Auto Focus (AF) → Auto Stigmation (AS), AS → AF) and the proposed method, with units for stigmator in percentage (%) and W. D. in micrometers (μm). Each sample was represented with a scatter plot, where the mean of error points was visualized using a bar plot, accompanied by error bars indicating ±1σ.

utilizes PyTorch[62] GPU version to offload intensive computations like convolutions to the GPU, thus enhancing processing efficiency.

A notable challenge in the practical application was the lengthy computational time associated with iterative reconstruction-based optimization, involving both inner deconvolution and outer compensation loops. To mitigate this, GPU acceleration was employed for fundamental operations such as general convolution and depthwise convolution using the PyTorch library. This approach drastically reduced processing times from approximately 1 minute per beam parameter set on a CPU to under 5 seconds on a GPU. This advancement in processing speed is particularly crucial in a field

like semiconductor manufacturing, where throughput substantially influences production capacity.

### Electron beam simulation in details

To establish the relationship between beam parameters and beam kernels, a Python-based Monte Carlo simulation was developed. This simulation differs considerably from traditional ray tracing used in optics. In optics, refraction through glass lenses is the primary concern, whereas in our simulation, we focus on the continuous path variations of electrons due to interactions with electromagnetic lenses and a stigmator. The simulation

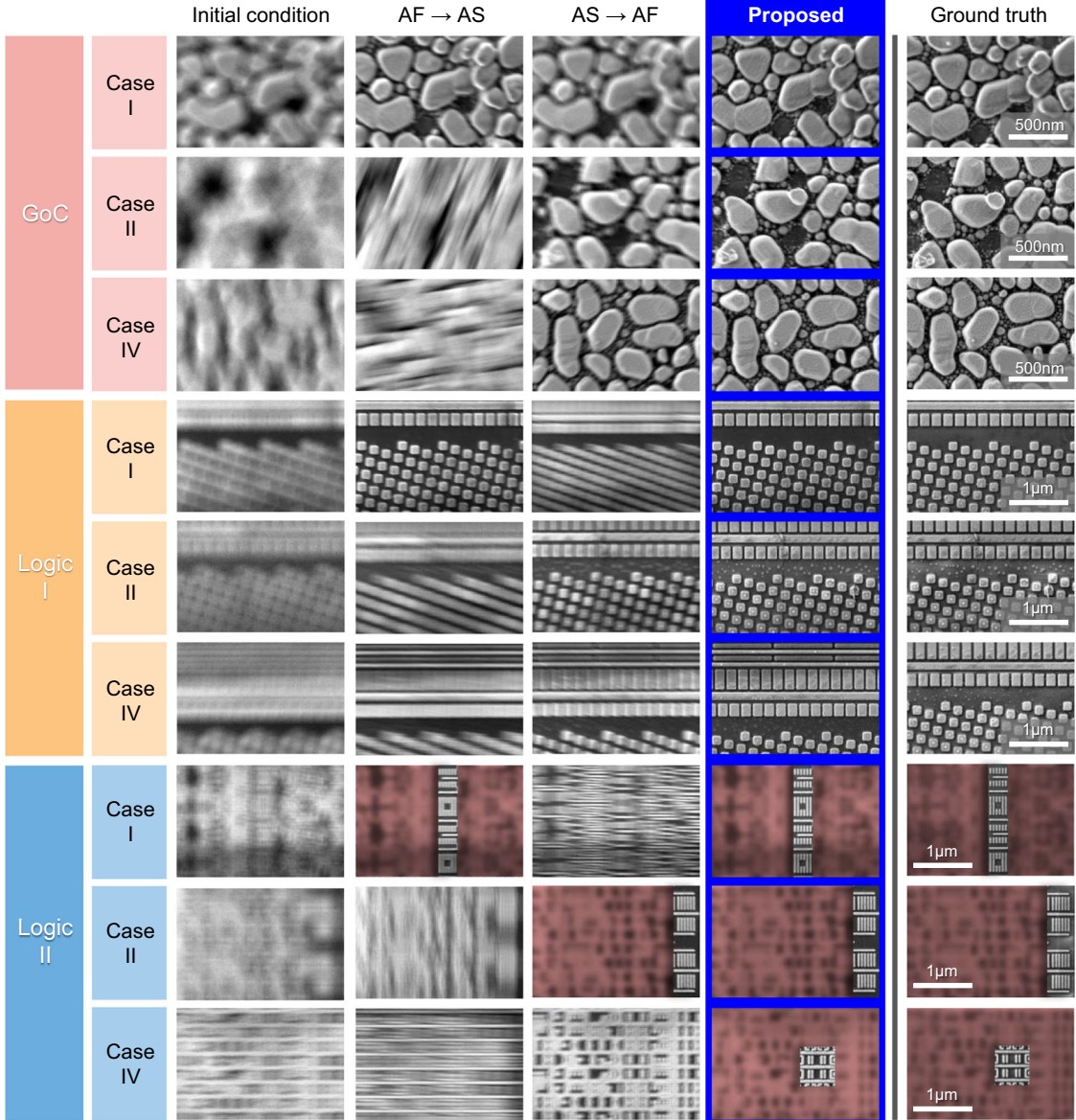

**Fig. 9 | Comparative analysis of result images using different methods.** This figure showcases optimization results for GoC, Logic I, and Logic II samples under various conditions. Displayed in sequence are images from the initial conditions, followed by results from the Auto Focus (AF) → Auto Stigmation (AS) method, AS → AF method, the proposed method, and the ground truth. The cases tested include focus misalignment (Case I), astigmatism (Case II), and combined misalignment of focus, stigmator, and aperture (Case IV). In the Logic II sample, certain areas have been blurred and colored red for security purposes. The scale bars are not provided for semiconductor images also for security reasons. The images reveal the effectiveness of the proposed algorithm in maintaining image quality comparable to the GT, even in scenarios where all parameters are misaligned.

initiates at the electron gun tip, where the incidence and azimuth angles of electrons are randomly set within a specified range, creating electron rays. The acceleration voltage is fixed at 10 keV, determining the electrons' initial velocity. The elevation angle from the gun varies randomly between 0 to 1.5 degrees, and the azimuth angle ranges from 0 to 360 degrees. It is assumed that the electrons do not experience any energy spread. Each electron ray is defined by a position vector <x, y, z> and a velocity vector <dx, dy, dz> . As the rays propagate in the positive z-direction, they interact sequentially with the condenser lens, aperture, second condenser lens, stigmator, and objective lens. These interactions, influenced by the modules' magnetic fields, lead to changes in the beam's collimation, position, and velocity. The Lorentz force, described by the formula $\mathbf{F} = q(\mathbf{E} + \mathbf{v} \times \mathbf{B})$, plays a crucial role in these trajectory modifications. In this formula, $\mathbf{F}$ represents the Lorentz force, $\mathbf{E}$ is the electric field, $\mathbf{B}$ is the magnetic field, and $\mathbf{v}$ is the electron's velocity. The electric field exerts a force independently, while the magnetic field impacts the electron's path perpendicularly.

For magnetic field computations, we utilized the open-source Python library MagpyLib[63]. The magnetic field is calculated volumetrically, varying with each beam parameter. These volumetric data are then incorporated into the ray simulation space. The volume grids for the lenses and stigmator are set at 10 ×10 × 20, with dimensions of 0.6 cm × 0.6 cm × 1.2 cm. As an electron ray moves through this volumetric space, the Lorentz force at each voxel is calculated, leading to updates in acceleration, velocity, and coordinates for the subsequent voxel. This methodology results in a continuous alteration of the electron trajectory within the volume. Regarding elements like the aperture, its radius is fixed, and only the x and y alignments are adjustable. The aperture's function is to permit the passage of rays within its circular boundary while blocking those outside it.

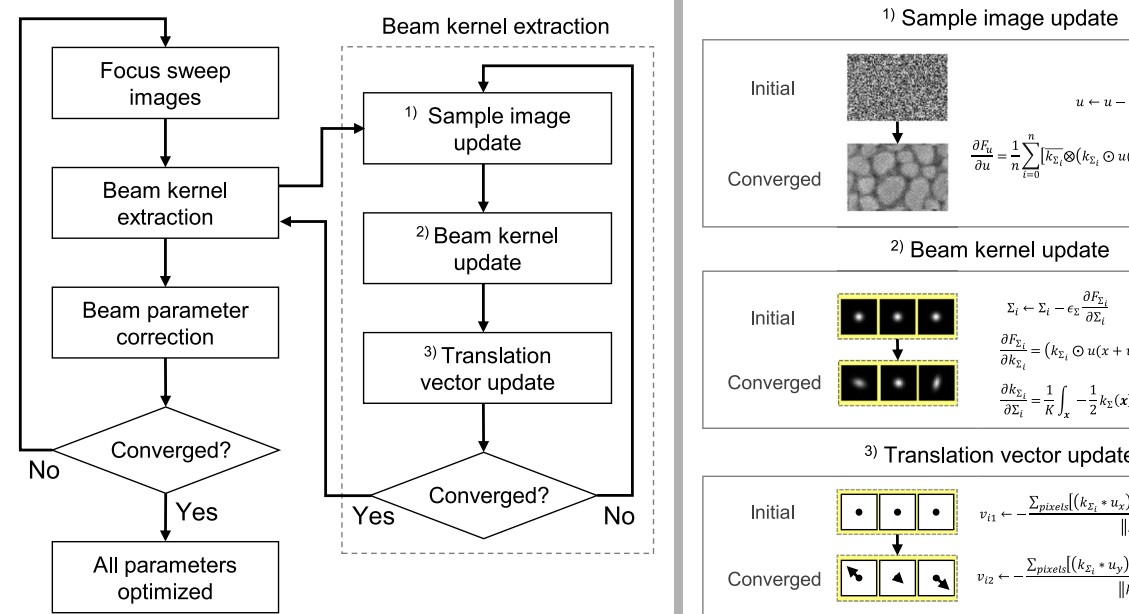

**Fig. 10 | Overall process of automatic beam correction method.** Illustrating the comprehensive steps from focus sweep Scanning Electron Microscopy (SEM) images to the final beam parameter correction, this figure details the methodical progression involving beam kernel extraction and iterative updates. It outlines the sequential updates of the sample surface, beam kernels, and translation vector. From the extracted beam kernels, key features related to stigmator, focus, and aperture-align are identified. Utilizing these features, the method derives compensatory vectors and corresponding adjustment rules, leading to the final correction of beam parameters.

## Overall optimization process of automatic beam optimization

The optimization of beam parameters, as described in Results, involves a repetitive process that includes beam kernel extraction and feature-based beam optimization. Each iteration of this process constitutes a single update of the compensatory vector towards reaching the optimal point. The challenge lies in determining not just the direction of compensation based on the current state of the parameters but also the magnitude of the necessary adjustments. The entire sequence from the initial to the converged states of all beam parameters is depicted in Fig. 10.

The process begins by acquiring a set of focus sweep images while altering the focus under the current conditions. One constraint here is that there must be an on-focus point within the focus sweep range. If there is no on-focus point within the sweep range and all images are excessively blurred, it becomes challenging to estimate a single sharp image through blind deconvolution. Therefore, it is advisable to start with a large sweep range (e.g. 100 μm) to ensure that the on-focus point falls within it, and then gradually narrow down the sweep range step by step. The acquired sweep images are then processed through the blind deconvolution method to yield a single surface image and an array of estimated beam kernels. This blind deconvolution phase involves an iterative update process[57], beginning with a random initialization of the surface image and kernel images, continuing until convergence is achieved. Once the kernels have converged, marking the completion of the inner loop, they are then utilized for beam parameter correction. Features extracted from each kernel are aligned linearly in their respective feature spaces for stigmator, working distance, and aperture-align. Based on these features, compensatory vectors containing both magnitude and phase information are computed. These vectors are instrumental in updating the beam parameters. The process of beam kernel extraction and beam parameter correction is repeated in cycles until the beam parameters converge, signifying the completion of the outer loop. The convergence criterion is based on the absolute variation of all beam parameter values falling below a specific threshold, ensuring optimal parameter adjustment. Please refer to the Supplementary

Movie 1 for footage of the entire optimization process running on the SEM equipment.

## Detailed view on deconvolution process for extracting beam kernels

The extraction of beam kernels through deconvolution involves an objective function as mathematically expressed in Eq. (1). This function consists of two primary components: the fidelity term and the regularization term. The fidelity term ensures that when the sharp surface image u, translated by vector v, is convolved with the beam kernel k, it closely matches the acquired SEM image f. This fit is averaged over multiple images obtained through focus sweeps. While each fidelity term is independent, they all share the common surface image u, underpinning the assumption that despite focus-induced kernel shape variations, the original surface image remains constant. The regularization term, known as the total variation term, reinforces the premise that the original surface image is inherently sharp before convolution. The regularization's weighting is controlled by the parameter λ. The optimization objective is to determine the surface image u, the Gaussian kernels' covariance matrices $\Sigma_1$ to $\Sigma_N$, and the displacement vectors $v_1$ to $v_N$ that collectively minimize the objective function. Gradient descent[64] is used for this optimization, differentiating the objective function with respect to each variable.

In the case of the beam kernel as shown in (2), it assumes a Gaussian shape, so it updates the 2×2 covariance matrix of Gaussian. Therefore, it takes an approach to differentiate the objective function with respect to the covariance matrix. Since the objective function is a function of the kernel image, k, the chain rule is used to differentiate it with respect to $k_i$ and then multiply it by the differentiation of $k_i$ with respect to $\Sigma_i$. The convolution notations $\odot$ and $\otimes$ denotes full type and valid type convolution respectively. When updating $\Sigma_i$ using gradient descent, the beam kernel also gets updated. Covariance matrix starts from an isotropic Gaussian shape, which is an identity matrix, and gradually updates to become an elliptical Gaussian with specific orientation and size as the updates progress. As discussed in Results, it is possible to extract features related to focus and stigmator from

the covariance matrix.

$$\Sigma_i \leftarrow \Sigma_i - \epsilon_\Sigma \frac{\partial F_{\Sigma_i}}{\partial \Sigma_i}, \frac{\partial F_{\Sigma_i}}{\partial \Sigma_i} = \frac{\partial F_{\Sigma_i}}{\partial k_{\Sigma_i}} \cdot \frac{\partial k_{\Sigma_i}}{\partial \Sigma_i},$$

$$k_{\Sigma_i}(\boldsymbol{x}) = \frac{1}{\sqrt{(2\pi)^2 |\Sigma_i|}} \exp\left(-\frac{1}{2}\boldsymbol{x}^T \Sigma_i^{-1}\boldsymbol{x}\right)$$

$$\frac{\partial F_{\Sigma_i}}{\partial k_{\Sigma_i}} = (k_{\Sigma_i} \odot u(x + v_i) - f_i) \otimes \overline{u(x + v_i)},$$

$$\frac{\partial k_{\Sigma_i}}{\partial \Sigma_i} = \frac{1}{K}\int_{\boldsymbol{x}} -\frac{1}{2}k_{\Sigma_i}(\boldsymbol{x})(\Sigma_i^{-1} - \Sigma_i^{-1}\boldsymbol{x}\boldsymbol{x}^T\Sigma_i^{-1})d\boldsymbol{x}$$

(2)

The $v$ mentioned in (3) is a vector related to translation. We opted to base it on the image rather than the kernel, as determining translation with respect to the image makes it easy to optimize. For the $u(x + v_i)$ in (3), it can be approximately expanded as $u(x) + \nabla u(x) \cdot v_i$. By solving Eq. (3), it can be determined how each image $f_i$ has shifted from $u$ along the direction $v_i$. The set of parameters $v_i$ is used as a feature in itself to control aperture alignment.

$$v_i \leftarrow v_i - \epsilon_v \frac{\partial F_{v_i}}{\partial v_i},$$

$$F_{v_i} = \frac{1}{N}\sum_{i=1}^{n}\|k_{\Sigma_i} * u(x + v_i) - f_i\|_2^2 \approx \sum_{i=1}^{n}\|k_{\Sigma_i} * (u + \nabla u \cdot v_i) - f_i\|_2^2$$

$$\frac{\partial F_{v_i}}{\partial v_i} = \sum_{pixel}(k_{\Sigma_i} * (u + \nabla u \cdot v_i) - f_i) \cdot (k_{\Sigma_i} * u_x),$$

(3)

Lastly, the surface image, $u$, can be updated as in (4). As can be inferred from the differentiation equation, $u$ is also related to the total variation term. Here, u starts as a random noise image following an initial uniform distribution and gradually takes on the form of a surface image as iterations progress. While $u$ is not directly used in the actual correction, it is employed as a key intermediate component for updating $v$ and $k$. Please refer to Supplementary Algorithm I in the Supplementary Information for the pseudo-code of the entire blind deconvolution algorithm.

$$u \leftarrow u - \epsilon_u \frac{\partial F_u}{\partial u}$$

$$\frac{\partial F_u}{\partial u} = \frac{1}{n}\sum_{i=0}^{n}[\overline{k_{\Sigma_i}} \otimes (k_{\Sigma_i} \odot u(x + v_i) - f_i)] - \lambda\nabla \cdot \frac{\nabla u}{|\nabla u|}$$

(4)

## Constraints in blind deconvolution

Blind deconvolution, being an underdetermined system with more unknown parameters than known inputs, requires the application of specific constraints to achieve reliable and meaningful solutions. In addition to the total variation constraint incorporated into the problem, several other strategic constraints were applied:

Firstly, the beam kernel shapes are fixed to Gaussian forms, simplifying the problem from estimating all pixels in a $21 \times 21$-sized kernel to estimating just four variables in a $2 \times 2$ covariance matrix. Due to the symmetry of the covariance matrix, effectively only three variables are required for estimation. Secondly, gradient clipping is used during the beam kernel update process. This step is crucial to prevent the covariance matrix gradient values from becoming excessively large, ensuring stability in the estimation. Thirdly, covariance clipping is implemented to prevent any individual element within the covariance matrix from becoming too large, which would result in an overly distorted ellipse. If any value exceeds a specified threshold, the entire matrix is proportionally scaled down. Finally, the update speed parameters $\epsilon_\Sigma$, $\epsilon_v$ and $\epsilon_u$ in Eqs. (2), (3), and (4) are subjected to gradual decay as iterations progress. This approach helps to reduce oscillations and foster stable convergence.

## Convergence judgment

For judging the convergence of the algorithm, the sum of the absolute changes in stigmator x, stigmator y, aperture-align x, and aperture-align y is

monitored. If this sum falls below a predetermined threshold for three consecutive iterations, the algorithm is considered to have converged. This criterion is based on the understanding that three consecutive iterations with minimal parameter variation signify stable convergence. It's important to note that the effectiveness of the algorithm is influenced by the scale of operation. The method is optimized for a magnification range between 20,000x and 30,000x. At very high magnifications, the kernel size might exceed the pixel size of a $21 \times 21$ kernel, introducing increased estimation uncertainty. Therefore, the algorithm's reliability and accuracy are best within this specified magnification range.

## Data availability

The data that support the findings of this study are available from the corresponding author upon reasonable request. Please note that the semiconductor-related data is not available due to security reasons.

## Code availability

We provide pseudo code at a level easily reproducible through Supplementary Information. Please note that the raw source code is not available due to company policy. For the reproducibility of this paper, both the SEM controllable via API and the corresponding function codes for controlling this API are required. However, since these vary for each equipment, individual implementations are necessary.

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

## Acknowledgements

This research was supported by the Mechatronics Research, Samsung Electronics Co., Ltd.

## Author contributions

Y.C and J.C conceived the original idea and developed the system and theory. Y.C, J.C and J.P developed the automatic beam optimization method. Y.C, J.C, and S.S developed the system software and the algorithm. J.C and J.P conducted experiment and analyzed the data. Y.C and J.P wrote the manuscript with support from T.J and M.L. Y.C and K.K were in charge of the overall direction and planning. J.W, S.J, J.L, Y.H, J.K, J.Y, H.C, H.P and S.S reviewed manuscript. K.K supervised the project. All authors provided critical feedback and helped shape the research.

## Competing interests

The authors declare no competing interests.
