## [Peer Review File · Communications Engineering]

Reviewers' comments:

Reviewer #1 (Remarks to the Author):

The manuscript describes a novel method to measure and optimize five parameters which are needed to align a scanning electron microscope (focus, astigmatism x, astigmatism y, aperture x and aperture y). The proposed algorithm uses a blind deconvolution approach where it is assumed that the image formation in a SEM can be described by the convolution of a point spread function (psf) with a unknown sample. The psf is described by a gaussian kernel, this assumption is validated using electron beam simulation in a typical SEM. Moreover, they show how the algorithm converges to the right solution for different starting positions when the different parameters are not at the optimal value. They also compare the method to other more conventional auto-focus and auto-stigmatation algorithms and show that it is more robust and converges to the estimated optimal settings determined by an expert user.

I believe that the work shown in this manuscript is novel and valuable to the community when following comments are answered and taken care of in the manuscript. The main concern is the comparison between the AF-AS method and the proposed method for Case4. This comparison is not fair because the other methods are not able to correct for this misalignment. In the manuscript it is emphasized that for this case, the proposed algorithm outperforms the other methods which is trivial since they cannot correct for this misalignment (Line 286). Perhaps, the authors could rewrite this part to clarify that the other algorithms cannot correct the aperture position which is disadvantage of the other methods and remove the quantitative comparison for Case4 (Figure 8) while keeping the visual aid of Figure 9 which then shows the necessity of correcting the aperture position.

Other minor comments are found below:

- Scalebars are missing from some images (figure 1 and 9), this information is useful for the reader to know which fields of view are accessible with the method.
- Line 59: Typo on astigmatism
- Line 235: How is the balancing factor λ chosen for the blind deconvolution algorithm (Eq 1)?
- Line 234: You describe that astigmatism can be identified when going from under to over focus. However how can you be sure that the focus sweep that you use goes through this circular form? Does the algorithm still converge if the probe does not go from under to over focus but remains under or over focused for the focus sweep.
- How is the error defined for both defocus and stigmators? For instance, the stigmator has an x and y component, how do you define the error on this?
- Figure 8. The box plots are not easily visible. Perhaps the authors could replace the box plots with a table showing the average value and standard deviation of defocus, stigmator x and stigmator y? Other possibilities would be to show the histograms, 1D histograms for defocus and 2D histograms for stigmator from which the bias and precision could be identified.
- Figure 8: How many tests are performed and are the same starting positions used for each method.
- Have you tested time for manual alignment versus the automatic? Is the convergence speed faster, slower or comparable to an expert user?
- Figure 9: Logic 2 seems to have a different height between two parts. Hence you can focus on the small block or the large part? How can you define a good focus point since both are equally valid.
- I would strongly suggest sharing the source code of the blind deconvolution algorithm with an example on the GoC sample since this would help other scientist implement this method in their own labs and it increases credibility towards the proposed method.

- Can you comment on the use of this method for higher magnifications? Is it possible to use it or are there reasons why it would/could fail?

Reviewer #2 (Remarks to the Author):

The organization and logical development of the paper is clear and understandable. On the other hand, for those who understand SEM, there are a few words in the paper that may be of concern. They are listed below.

Distortion (e.g. line 56, 58, etc.)

"Distortion" usually refers to local distortion within a screen, and should be distinguished from the blurring of the entire screen shown in the paper. I think it would be better to change the word.

Modern (Line63)

I think it is more common to use "advanced" than "modern".

Input/Output (Line65)

If you are suggesting operation with CD-SEM, I think Load/Unload or Loading/Unloading is appropriate.

Clean Surface Image/Clean Image (Line201,202,)

In the SEM world, a clean image is an image taken in a contamination-free or damage-free state. I thought this point might be at odds with what the author is trying to say. What about Correct Image, for example?

Strike (line 248)

Strike gives the image of an electron beam striking a sample. Since the electron beam scans the surface of the sample, I think irradiation would be a more accurate description.

Reviewer #3 (Remarks to the Author):

In the work by Cho et al, they describe an Automatic Beam Optimization Method for SEM. They achieve this goal by changing the beam focus and acquiring images. Assuming that drift and other problems are not significant, the translation gives an indication of the aperture mis-alignment and the ellipticity of the deconvolved kernel gives the focus and astigmatism. Taking a sequence of a few images allows them to obtain estimates for the controls needed to correct the images. The novel points are that they do a monte-carlo simulation of the electron trajectories and that they use a gradient descent method to estimate the kernel. Section 4.2: It's interesting that they do this simulation. In some ways, I think they could more simply parameterize the problem, but this step probably gives them some extra insight. Overall it is well written and reasonably well presented. I suggest accepting it with perhaps a few minor changes.

Minor points/typos

Page 3: line 59, Unnecessary "b".

Page 3: Probably I would say "sub-nm" rather than angstrom in this case.

Fig 2: I prefer not to call 2 crossed quadrupoles an octupole, as the symmetry is different.

2.1: I feel most of this, and some of the next, section could be replaced with saying that to a reasonably good approximation, the SEM image can be represented as a convolution.

Page 14: line 239, I think "We feature" is an odd way of saying something.

Line 252: "PCA determines" - they need to describe this step a bit more. Although it makes more sense after reading the Methods section, I wasn't quite clear what they are doing.

Add some scale bars on images.

The references are good and reasonably comprehensive.

Possibly where they mention TEM or STEM (refs 18 or 35 or 45,46,47) they might want to cite also:

O.L. Krivanek, G.Y. Fan, Scanning Microscopy 6 (Suppl.) (1992) 105

H. Sawada, et al., Ultramicroscopy 108 (2008) 1467.

O.L. Krivanek, N. Dellby, A.R. Lupini, Ultramicroscopy 78 (1999) 1.

F. Zemlin, et al., Ultramicroscopy 3 (1978) 49

The reason to add those is that there's a small thing I disagree with in, "Other methods, such as those utilizing additional information like beam tilt, have been investigated but do not offer significant performance advantages" – they could be clearer that they mean for this specific application. The other point I might add is that in TEM/STEM there are additional aberrations to worry about and there tilt (or otherwise looking at the angular dependence more closely) might provide extra information.

Responses to the Reviewer Comments

Submission Manuscript [COMMSENG-24-0003A]

Title: Automatic Beam Optimization Method for Scanning Electron Microscopy based on Electron Beam Kernel Estimation

Reviewer Comments:

Reviewer #1

The manuscript describes a novel method to measure and optimize five parameters which are needed to align a scanning electron microscope (focus, astigmatism x, astigmatism y, aperture x and aperture y). The proposed algorithm uses a blind deconvolution approach where it is assumed that the image formation in a SEM can be described by the convolution of a point spread function (psf) with a unknown sample. The psf is described by a gaussian kernel, this assumption is validated using electron beam simulation in a typical SEM. Moreover, they show how the algorithm converges to the right solution for different starting positions when the different parameters are not at the optimal value. They also compare the method to other more conventional auto-focus and auto-stigmatism algorithms and show that it is more robust and converges to the estimated optimal settings determined by an expert user.

1. I believe that the work shown in this manuscript is novel and valuable to the community when following comments are answered and taken care of in the manuscript. The main concern is the comparison between the AF-AS method and the proposed method for Case4. This comparison is not fair because the other methods are not able to correct for this misalignment. In the manuscript it is emphasized that for this case, the proposed algorithm outperforms the other methods which is trivial since they cannot correct for this misalignment (Line 286). Perhaps, the authors could rewrite this part to clarify that the other algorithms cannot correct the aperture position which is disadvantage of the other methods and remove the quantitative comparison for Case4 (Figure 8) while keeping the visual aid of Figure 9 which then shows the necessity of correcting the aperture position.

■ The authors express gratitude for the reviewer's comments and agree with the reviewer's opinion that the comparison with the existing method for Case 4, which involves aperture misalignment, is not fair. Therefore, as suggested by the reviewer, Case 4 has been removed from Fig. 8, and instead, emphasis has been placed on the ability to align the aperture in the text. Furthermore, as mentioned in the minor comments, it was noted that the box plot in Fig. 8 was not easily visible. Therefore, the authors have replaced it with a simple yet clear scatter plot format. The authors have represented the mean error values in a bar chart format for ease of interpretation and have also explicitly displayed the values.

(Original Figure)

(Revised Figure)

a CASE I – Focus misaligned

b CASE II – Stigmator misaligned

c CASE III – Focus, Stigmator misaligned

Fig. 8 | Accuracy performance evaluation.

Performance evaluations for three cases (**a** CASE I: focus misaligned, **b** CASE II: stigmator misaligned, and **c** CASE III: both focus and stigmator misaligned) across three samples (GoC, Logic I, and II). Error dispersion for stigmator and working distance (W. D.) is shown, along with comparisons made between sharpness based methods (AF→AS, AS→AF) and the proposed method, with units for stigmator in percentage (%) and W. D. in micrometers (μm). Each sample was represented with a jittered scatter plot, where the mean of error points was visualized using a bar plot, accompanied by error bars indicating $\pm 1\sigma$.

Line 275-288:

The proposed method's performance was compared with conventional sharpness-based methods using auto-correlation^{31,32}. These methods were applied as auto-focus followed by auto-stigmatism (AF→AS) and auto-stigmatism followed by auto-focus (AS→AF). Experiments were conducted on Gold on Carbon (GoC) and two semiconductor logic device samples under four conditions: out-of-focus (Case I), astigmatism (Case II), out-of-focus with astigmatism (Case III), and out-of-focus with astigmatism and aperture-misalign (Case IV). Figure 8 presents a jittered scatter plot of error dispersion relative to the GT. In **a** (Case I), AF→AS outperforms between sharpness-based methods, while in **b** (Case II), AS→AF is better. This is because if only one of focus or stigmator is misaligned, aligning that parameter first can enable the system to operate accurately. However, in situations with both stigmator and focus misaligned, these methods diverge from the optimal point. The

proposed method converges near the optimal point not only in simple cases like **a** and **b**, but even in more complex scenarios like **c** (Cases III). When aperture-align x, y deviates from the optimum points, as well as the preceding three parameters (CASE IV), conventional correction methods may be ineffective. However, all five parameters can be accurately adjusted simultaneously using the proposed method. It shows significantly lower error levels (focus: $1.00\mu\text{m}$, stigmator: 0.30% , aperture-align: 0.79%) for all GoC, Logic I, and II samples.

2. Other minor comments are found below:

2-1. Scalebars are missing from some images (figure 1 and 9), this information is useful for the reader to know which fields of view are accessible with the method.

■ As suggested, scale bars have been added to Fig. 1 and Fig. 9. However, due to the sensitive nature of scale information in semiconductor samples, scale bars have been omitted for these samples.

(Revised Fig. 1)

(Revised Fig. 9)

2-2. Line 59: Typo on astigmatism

■ The authors apologized typos in the manuscript, and have changed typo in the manuscript.

(bastigmatism -> astigmatism)

2-3. Line 235: How is the balancing factor lambda chosen for the blind deconvolution algorithm (Eq 1)?

■ Lambda is a parameter responsible for balancing between the fidelity term and the regularization term. It determines the weight given to the regularization term, which penalizes sparse gradients in the estimated surface image, indicating how sharp the estimated image should be. The authors conducted a typical parameter range search and ultimately selected a value of 0.001 for lambda. This chosen lambda value is robust, so it was used consistently throughout the entire experiment.

2-4. Line234: You describe that astigmatism can be identified when going from under to over focus. However how can you be sure that the focus sweep that you use goes through this circular form? Does the algorithm still converge if the probe does not go from under to over focus but remains under or over focused for the focus sweep.

■ The authors appreciate the reviewer's comments on pointing out this important aspect. The on-focus point should lie between under-focus and over-focus to function as intended. If all images are excessively blurred and the shape is unrecognizable, it becomes challenging to extract a sharp surface image. This can be considered as the constraint of the proposed method. Considering the H/W error budget of the SEM equipment we use in our experiments, it appears that the on-focus plane falls well within the initial sweep range of 100 micrometers. If the on-focus plane were not within the sweep range, it would lead to the issues mentioned earlier. However, we believe that the sweep range could sufficiently covers a wide range. The manuscript has been revised following the comments.

Line 370-376:

The process begins by acquiring a set of focus sweep images while altering the focus under the current conditions. One constraint here is that there must be an on-focus point within the focus sweep range. If there is no on-focus point within the sweep range and all images are excessively blurred, it becomes challenging to estimate a single sharp image through blind deconvolution. Therefore, it is advisable to start with a large sweep range (e.g. 100um) to ensure that the on-focus point falls within it, and then gradually narrow down the sweep range step by step. The acquired sweep images are then processed through the blind deconvolution method to yield a single clean image and an array of estimated beam kernels.

2-5. How is the error defined for both defocus and stigmators? For instance, the stigmator has an x and y component, how do you define the error on this?

■ Each error was calculated as the Euclidean distance between the ground truth (GT) and the estimated value. Multiple tests were conducted for each sample, and the Euclidean distance errors obtained from these tests were averaged to represent accuracy.

2-6. Figure 8. The box plots are not easily visible. Perhaps the authors could replace the box plots with a table showing the average value and standard deviation of defocus, stigmator x and stigmator y? Other possibilities would be to show the histograms, 1D histograms for defocus and 2D histograms for stigmator from which the bias and precision could be identified.

■ The reviewer's feedback has been incorporated, and Fig. 8 has been improved accordingly. The updated figure can be found in the response to comment 1.

2-7. Figure 8: How many tests are performed and are the same starting positions used for each method.

■ For each algorithm and case, the authors conducted 8 tests, with the initial points for the 8 tests as follows for the Stigmator: (-3%, -3%), (-3%, 0%), (-3%, 3%), (0%, -3%), (0%, 3%), (3%, -3%), (3%, 0%), and (3%, 3%), relative to the ground truth (GT). Additionally, the working distance (WD) was initialized within a range of up to 6 micrometers from the GT.

2-8. Have you tested time for manual alignment versus the automatic? Is the convergence speed faster, slower or comparable to an expert user?

■ The time taken by an expert to manually adjust five parameters is empirically around 5 minutes. Our algorithm, however, is faster, typically taking around 2 minutes or less compared to manual adjustment. When adjusting manually, a person typically moves the focus until the image is visible, then sequentially adjusts the stigmator x, stigmator y, aperture x, and aperture y while checking the image. Afterward, they need to repeat the process of adjusting all parameters starting from the focus again. The proposed method involves decoupling the correlation among the five parameters and adjusting them simultaneously, which has enhanced speed compared to manual adjustment.

2-9. Figure 9: Logic 2 seems to have a different height between two parts. Hence you can focus on the small block or the large part? How can you define a good focus point since both are

equally valid.

■ The blurred parts in Logic 2 is excessively blurred not due to differences in height but for security reasons. Please refer to the part mentioned in “In the Logic II sample, certain areas have been blurred for security purposes” in the caption. Following the reviewer's comment, the colors have also been brightened to ensure better recognition of this aspect and to mitigate any potential misunderstanding.

2-10. I would strongly suggest sharing the source code of the blind deconvolution algorithm with an example on the GoC sample since this would help other scientist implement this method in their own labs and it increases credibility towards the proposed method.

■ The authors have added detailed pseudo code as supplementary information. Additionally, all images except for semiconductor images can be provided upon request. Unfortunately, due to company policies, sharing raw code is not feasible. However, the authors believe that with the pseudo code provided, it should be relatively straightforward to reproduce the results. Other additional information, excluding raw code, can be shared upon request, subject to company approval. However, please note that for reproduction purposes, access to a SEM facility with a publicly available API and specific code for operation (which varies depending on the equipment) will be necessary.

2-11. Can you comment on the use of this method for higher magnifications? Is it possible to use it or are there reasons why it would/could fail?

■ Our method operates independently of patterns, so it is theoretically feasible to work at higher magnifications as well. However, at higher magnifications, larger kernel size (pixel-countwise) are required, which necessitates larger computations. Currently, we are using a 21 px. x 21 px. kernel size, but larger kernel sizes are required at higher magnifications. If sacrificing some accuracy is acceptable, resizing the image and kernel for computation could be a viable approach.

Reviewer #2

The organization and logical development of the paper is clear and understandable. On the other hand, for those who understand SEM, there are a few words in the paper that may be of concern.

1. They are listed below.

1-1. Distortion (e.g. line 56, 58, etc.)

"Distortion" usually refers to local distortion within a screen, and should be distinguished from the blurring of the entire screen shown in the paper. I think it would be better to change the word.

■ The authors appreciate reviewer's comment for clarifying the meaning of the terms. The authors had used the word "distort" in various parts of the paper, but as the reviewer mentioned, there may be some potential for misunderstanding. Therefore, the authors have replaced "distortion" related to the image with "blurriness" and changed the word "distortion" related to the kernel to "deformation" to make the meanings clearer.

Line 53: This precision enables the consistent acquisition of sharp and correct images.

Line 54: It displays SEM images under different conditions: **a** shows a sharp image without any blurriness, **b** an image with out-of-focus but without astigmatism, **c** an image on-focus but with astigmatism, and **d** an image out-of-focus and with astigmatism.

Line 56: These examples highlight how deviations from optimal conditions, such as out-of-focus or stigmator aberration, lead to blurred or directionally **blurred** images. Such image quality issues can result in measurement inaccuracies during CD measurement and cause false positives or failures in defect inspection.

Line 91: While these methods have their merits, they are vulnerable to noise in an image and do not measure scores robustly for severely **blurred** images.

Line 148: The pursuit of optimal quality in SEM imaging is intrinsically linked to achieving the smallest and least **deformed** beam kernel. A smaller beam kernel enhances resolution, while **deformation** in a beam kernel leads to blurring in specific image directions.

Line 185: In summary, incorrect states of the stigmator and aperture-align lead to beam kernel **deformation**, thereby **blurring** SEM images. Consequently, our method ascertains the beam kernel state, enabling us to reverse engineer these parameters to yield high-resolution images.

1-2. Modern (Line63)

I think it is more common to use "advanced" than "modern".

■ The authors agreed with the reviewer's suggestion that "advanced" is more suitable than "modern" in the sentence, and accordingly, we revised the manuscript as follows.

Line 59: Given the trend toward sub-nanometer scale patterning precision in **advanced** device processes, the need for technology that can precisely optimize the electron beam is paramount.

1-3. Input/Output (Line65)

If you are suggesting operation with CD-SEM, I think Load/Unload or Loading/Unloading is appropriate.

■ The authors agree with the reviewer's suggestion that "loading/unloading" is more appropriate than "input/output." They have accordingly revised the manuscript as follows. Please review the changes below.

Line 61: Moreover, this optimization may need to be performed frequently due to changes in the beam over time, different beam conditions used, and the **loading/unloading** of wafers.

1-4. Clean Surface Image/Clean Image (Line201,202,)

In the SEM world, a clean image is an image taken in a contamination-free or damage-free state. I thought this point might be at odds with what the author is trying to say. What about Correct Image, for example?

■ As per the reviewer's suggestion, the authors have replaced the somewhat ambiguous term "clean"

with "correct" or "correct surface." Please review the revised manuscript below.

Line 53: This precision enables the consistent acquisition of sharp and **correct** images.
Line 197: Given that only SEM images are available, we formulated a mathematical model using blind deconvolution techniques to estimate both the **surface** image and the kernel when both are unknown.
Line 204: Our constraints include the assumption that focus sweep images share the same **surface** image but with different kernels, the presumption that the **correct surface** image is sharp (TV regularization), and the beam kernel has Gaussian shape.
Line 224: This technique facilitates the extraction of beam kernels for each working distance, along with a common **surface** image shared by sweep images. The characteristics of these kernels, such as size, shape, and displacement direction, are closely linked to the beam parameters. Drawing on these characteristics, we have developed a method to correct key parameters, namely the stigmator, aperture alignment, and focus, for acquiring a **correct** image. Figure 6 illustrates the process of extracting compensatory directions for these key parameters from the acquired beam kernels.
Line 374: The acquired sweep images are then processed through the blind deconvolution method to yield a single **surface** image and an array of estimated beam kernels. This blind deconvolution phase involves an iterative update process⁵³, beginning with a random initialization of the **surface** image and kernel images, continuing until convergence is achieved.

1-5. Strike (line 248)

Strike gives the image of an electron beam striking a sample. Since the electron beam scans the surface of the sample, I think irradiation would be a more accurate description.

■ The authors would like to express gratitude to the reviewer for the helpful suggestion. We agree that "irradiate" is a more appropriate term than "strike," and have made the following modification accordingly.

Line 244: For aperture misalignment, where the electron beam does not **irradiate** the sample vertically, we observe a specific shift in the beam kernel's position with focus change.

Reviewer #3

In the work by Cho et al, they describe an Automatic Beam Optimization Method for SEM.

They achieve this goal by changing the beam focus and acquiring images. Assuming that drift and other problems are not significant, the translation gives an indication of the aperture mis-alignment and the ellipticity of the deconvolved kernel gives the focus and astigmatism. Taking a sequence of a few images allows them to obtain estimates for the controls needed to correct the images.

The novel points are that they do a monte-carlo simulation of the electron trajectories and that they use a gradient descent method to estimate the kernel.

Section 4.2: It's interesting that they do this simulation. In some ways, I think they could more simply parameterize the problem, but this step probably gives them some extra insight.

Overall it is well written and reasonably well presented. I suggest accepting it with perhaps a few minor changes.

1. Minor points/typos

1-1. Page 3: line 59, Unnecessary "b".

■ The authors apologized typos in the manuscript, and have changed typo in the manuscript.
(bastigmatism -> astigmatism)

1-2. Page 3: Probably I would say "sub-nm" rather than angstrom in this case.

■ The authors agree with the reviewer's comment that "sub-nanometer" is more suitable than "angstrom." They have made the following change accordingly.

Given the trend toward sub-nanometer scale patterning precision in advanced device processes, the need for technology that can precisely optimize the electron beam is paramount.

1-3. Fig 2: I prefer not to call 2 crossed quadrupoles an octupole, as the symmetry is different.

■ The authors have incorporated the reviewer's comments into the manuscript. Following the comment, the word "octupole" has been removed from the label on the Stigmator figure in Figure 2.

(Original Figure)

(Revised Figure)

1-4. 2.1: I feel most of this, and some of the next, section could be replaced with saying that to a reasonably good approximation, the SEM image can be represented as a convolution.

■ The authors appreciate reviewer's insightful feedback on manuscript. The authors have revised Section 2.1 to provide a more concise explanation of the SEM imaging process, emphasizing its representation as a convolution. The updated section now offers a streamlined overview of the physical model of SEM image acquisition and highlights the significance of beam optimization and kernel estimation in achieving high-quality SEM images. The authors believe these revisions address your suggestion effectively and enhance the clarity and relevance of our manuscript.

2.1. Image acquisition in SEM: beam kernel – object convolution

In this study, we interpret the physical model of the SEM imaging process by employing a computational model of image convolution for the numerical analysis of electron beam control. Figure 2a illustrates the physical model of the image acquisition process in an SEM, demonstrating the influence of beam parameters on SEM image acquisition. This process can be divided into two phases: the formation of an electron beam within the column and the determination of each pixel's value for constructing a 2D image. The shape of the scanning electron beam is determined by various hardware devices installed within the column⁵⁴, such as the emitter, aperture, stigmator, and objective lens. The cross-sectional shape of the electron beam irradiated on the sample's surface (focused in a circular or elliptical shape) correlates with the volume of interaction with the sample material, and the intensity of signals, such as secondary electrons, determines each pixel's value in the image. In this physical model, the optimization problem of the electron beam arises as the shape of the beam cannot be directly measured but must be inferred through acquired images. The resulting signal, as the electron beam irradiates the surface area, manifests as a weighted sum according to the beam kernel's shape, filling each image pixel. Consequently, the beam kernel operates analogously to an optical blur kernel or a point spread function. Discretizing an acquired SEM image conceptualizes it as the convolution⁵⁵ of a surface pattern image with the beam kernel image. Figure 2b illustrates the computational model interpreting the SEM acquisition process as a convolution between these two images: a sample surface image and a beam kernel image. Modeling this process as a convolution problem enables the numerical analysis and optimization of the physical elements of the beam kernel, as detailed in the following section.

1-5. Page 14: line 239, I think "We feature" is an odd way of saying something.

■ The authors have incorporated the reviewer's comments into the manuscript. The authors have replaced the word 'feature' with 'characterize' to clarify the meaning.

We **characterize** these shape changes of the kernel from under-focus to over-focus.

1-6. Line 252: "PCA determines" - they need to describe this step a bit more. Although it makes more sense after reading the Methods section, I wasn't quite clear what they are doing.

■ Agreeing with the reviewer's comment, I have rephrased the following to make it clearer. The authors have explicitly stated in the manuscript that the direction of the principal components resulting from PCA corresponds to the direction of compensation vectors.

Line 248: The direction of the first principal component vector calculated by Principal Component Analysis (PCA)⁵⁴ determines the compensatory vectors' direction, and their magnitudes are proportional to the standard deviation of the translation vectors' Euclidean distance distribution.

1-7. Add some scale bars on images.

■ As suggested, scale bars have been added to Fig. 1 and Fig. 9. However, due to the sensitive nature of scale information in semiconductor samples, scale bars have been omitted for these samples.

(Revised Fig. 1)

(Revised Fig. 9)

Fig. 9 | Comparative analysis of result images using different methods. This figure showcases optimization results for GoC, Logic I, and Logic II samples under various conditions. Displayed in sequence are images from the initial conditions, followed by results from the AF→AS method, AS→AF method, the proposed method, and the ground truth. The cases tested include focus misalignment (Case I), astigmatism (Case II), and combined misalignment of focus, stigmator, and aperture (Case IV). In the Logic II sample, certain areas have been blurred and colored red for security purposes. **The scale bars are not provided for semiconductor images also for security reasons.** The images reveal the effectiveness of the proposed algorithm in maintaining image quality comparable to the GT, even in scenarios where all parameters are misaligned.

1-8. The references are good and reasonably comprehensive.

Possibly where they mention TEM or STEM (refs 18 or 35 or 45,46,47) they might want to cite also:

O.L. Krivanek, G.Y. Fan, Scanning Microscopy 6 (Suppl.) (1992) 105

H. Sawada, et al., Ultramicroscopy 108 (2008) 1467.

O.L. Krivanek, N. Dellby, A.R. Lupini, Ultramicroscopy 78 (1999) 1.

F. Zemlin, et al., Ultramicroscopy 3 (1978) 49

The reason to add those is that there's a small thing I disagree with in, "Other methods, such as those utilizing additional information like beam tilt, have been investigated but do not offer significant performance advantages" – they could be clearer that they mean for this specific application. The other point I might add is that in TEM/STEM there are additional aberrations to worry about and there tilt (or otherwise looking at the angular dependence more closely) might provide extra information.

■ The authors appreciate reviewers for suggesting excellent references. The authors have added all the suggested references and revised the parts in the manuscript where they are mentioned.

Line 550-556

48. Sawada, H. et al. Measurement method of aberration from ronchigram by autocorrelation function.

Ultramicroscopy 108, 1467–1475 (2008).

49. Krivanek, O. L., Dellby, N. & Lupini, A. R. Towards sub-Å electron beams. Ultramicroscopy 78, 1–11 (1999).

50. Zemlin, F., Weiss, K., Schiske, P., Kunath, W. & Herrmann, K.-H. Coma-free alignment of high resolution electron microscopes with the aid of optical diffractograms. Ultramicroscopy 3, 49–60 (1978).

51. Krivanek, O. L. & Fan, G. Y. Application of Slow-Scan Charge-Coupled Device (CCD) Cameras to On-Line Microscope Control. Scanning Microscopy 6 (1992)

Line 97-99: Other methods have been developed using TEM/STEM to target specific aberrations of a particular equipment and resolve them by utilizing additional features such as tilt. However, this is not a universally applicable technique for equipment like SEM.^{45,46,47,61,62,63,64}.

REVIEWERS' COMMENTS:

Reviewer #1 (Remarks to the Author):

The authors have answered all the questions, and incorporated it well in the manuscript. I have no further questions or comments and recommend the publication of this work

Reviewer #2 (Remarks to the Author):

I recognized that the points I pointed out in the last review have been corrected. I do not think there are any other points that need to be corrected by this paper.

Reviewer #3 (Remarks to the Author):

In the first round, all three referees liked the paper, but all suggested some minor improvements.

I read the detailed response to all of my points and found that they have addressed all of my queries well. I quickly read the response to the other reviewers and I think that the authors have also addressed those points. I recommend the work be accepted.

Responses to the Reviewer Comments

Submission Manuscript [COMMS-24-0003B]

Title: Automatic Beam Optimization Method for Scanning Electron Microscopy based on Electron Beam Kernel Estimation

REVIEWERS' COMMENTS:

Reviewer #1

The authors have answered all the questions, and incorporated it well in the manuscript. I have no further questions or comments and recommend the publication of this work

- The authors express their gratitude for the valuable comments from the reviewers.

Reviewer #2

I recognized that the points I pointed out in the last review have been corrected.

I do not think there are any other points that need to be corrected by this paper.

- The authors are thankful for the valuable feedback provided by the reviewers.

Reviewer #3

In the first round, all three referees liked the paper, but all suggested some minor improvements.

I read the detailed response to all of my points and found that they have addressed all of my queries well. I quickly read the response to the other reviewers and I think that the authors have also addressed those points. I recommend the work be accepted.

- Thanks to the comments from the reviewers, we were able to improve the manuscript further. We appreciate the positive feedback.